# The Predictive Role of Affectivity, Self-Esteem and Social Support in Depression and Anxiety in Children and Adolescents

**DOI:** 10.3390/ijerph17196984

**Published:** 2020-09-24

**Authors:** Wenceslao Peñate, Melissa González-Loyola, Cristian Oyanadel

**Affiliations:** 1Departamento de Psicología Clínica, Psicobiología y Metodología, Facultad de Psicología, Campus de Guajara, Universidad de La Laguna, 38200 Santa Cruz de Tenerife, Spain; 2Instituto Universitario de Neurociencia, Universidad de La Laguna, 38200 Santa Cruz de Tenerife, Spain; 3Departmento de Psicología, Facultad de Ciencias Sociales, Universidad de Concepción, 4030000 Concepción, Chile; meligonzalez@udec.cl (M.G.-L.); coyanadel@udec.cl (C.O.)

**Keywords:** depression, anxiety, affectivity, self-esteem, social support, adolescence

## Abstract

*Background:* This study analyzes the relationship between depression and anxiety levels and positive and negative affect, self-esteem, and perceived social support from family and friends in an early and middle adolescent sample. These are psychological variables that are often associated with the prediction of emotional disorders, especially depression. *Methods:* Participants (*N* = 467) were a representative sample of this group of adolescents and were recruited from schools in the city of Concepción, Chile. Part of the sample (*N* = 177) was assessed three additional times—at one-, two-, and four-month intervals. *Results:* Results showed a practical stability of all measures across the four intervals, with no significant differences between sexes. Anxiety and depression displayed a similar pattern of significant relationships with affectivity, self-esteem, and social support. Depression had a higher correlation coefficient (−0.47) with positive affect, and so did anxiety with negative affect (0.58). *Conclusions:* Taking into account 23 initial scores on affectivity, self-esteem, and social support in predicting both depression and anxiety scores at one-month, two-month, and four-month intervals, positive affect was present in three regression analyses, predicting depression scores; negative affect was present in anxiety scores. Results are discussed according to previous findings, as well as the tripartite model.

## 1. Introduction

Emotional disorders, which are mainly characterized by depression and anxiety, are a highly prevalent group of disorders in individuals’ life spans [1]. The prevalence of mental disorders among children and adolescents ranges between 13% and 14%, and depression (2.6%) and anxiety (6.5%) are some of the most frequent disorders [2]. Moreover, depression and anxiety are highly comorbid in childhood and adolescence, where they frequently share a single cluster [3]. In Chile, the prevalence of mental disorders in the child–adolescent population is high. In fact, data show a 38.3% prevalence rate of mental disorders in children and adolescents between 4 and 18 years old. Of these, 22.5% are considered to be severe. The most common mental disorders are disruptive ones, followed by anxiety disorders and depressive disorders [4].

The stability of anxiety and depression problems seems to be associated with age. In fact, anxiety disorders exhibit a U-shaped pattern, with a high prevalence during childhood, which decreases in early adolescence and increases during middle and late adolescence [5]. Depressive disorders have been found to progressively increase in the transition from childhood to late adolescence, showing a linear pattern that is more marked in females [6].

The impact of these disorders is considerable: according to the Disability-Adjusted Life Years (DALYs), depression and anxiety are among the 10 most prevalent diseases of young people from 10 to 24 years old [7], and depression has a particularly prominent impact in the DALYs of young females [8]. Besides, the onset of these problems in adolescence is a risk factor for the development of mental disorders during adulthood [8].

Many risk factors for emotional disorders have been proposed from several different points of view. In this regard, a vast group of variables has been listed, including biological, temperamental, psycho-social, socio-contextual, and environmental ones [9,10,11]. The specific impact of these variables and their differential participation in predicting depression and anxiety disorders are still the subject of research controversies [11].

Beyond biological, environmental, and temperamental variables, a group of psychosocial variables have consistently been found to predict depression in childhood and adolescence. Regarding psychological variables, affectivity has been found to be a predictor of emotional disorders, as pointed out by the tripartite model [12]. It is well-established that emotional expressions are more intense and unstable among young people, particularly in the first few years of adolescence [13]. The absence of adaptive emotional control resources and the use of maladaptive strategies to cope with these emotional levels have been associated with the onset of emotional disorders in adolescence [14]. According to emotional valence, positive affect has been negatively associated with depressive mood levels, and negative affect has been positively associated with both depression and anxiety levels [15,16,17].

Self-esteem has also been found to be a relevant psychological variable. Low self-esteem has been associated with anxiety [18] and depression [19]. This association has been postulated (i) as a reciprocal relationship [20]; (ii) as a vulnerability condition for emotional disorders, especially depression [21]; and (iii) as a consequence of depressive states, playing a symbiotic role because, in turn, low self-esteem increases depression levels [22]. By contrast, high self-esteem has shown to be a protective factor for the development of mental disorders [23].

As regards social-family factors, social interaction and social support—including perceived social support have been found to be closely related to anxiety and depression problems [24,25,26,27]. Inadequate relationships between parents and their adolescent children (e.g., such as if they perceive their parents are using emotional manipulation strategies) have been found to be associated with the development of emotional disorders in adolescents [28]; conversely, adequate peer relationships and family care have been found to be conditions that support suitable emotional development [24]. Indirectly, social support can also prevent emotional problems by promoting self-concept and self-esteem [29].

We considered the previous data and the importance of certain psychosocial factors in the stability of depression and anxiety levels in children and adolescents. Consequently, the present study aimed to address the inter- and intra-subject stability of depression and anxiety scores, considering affectivity, self-esteem, and social support levels and addressing these variables simultaneously. To explore the relationships between these constructs appropriately, we conducted a longitudinal study of a child-youth sample for four months, with four measurement time-points. We expected these four measurements to add support to a consolidated prediction model for depression and anxiety levels in the child-youth population.

## 2. Materials and Methods

### 2.1. Participants

An initial representative sample (0.95 confidence level) was recruited from eight schools in the city of Concepción, Chile. The sample was stratified, considering the three types of schools present in Chile (i.e., municipal, private subsidized, and private non-subsidized). In these schools (three municipal (public) schools (37.90%), three private subsidized (29.12%), and two private non-subsidized), all students from fourth to eighth grade were invited to participate, and a total of 467 students (50.75% females, with ages ranging from 8 to 16 years) accepted to participate and were present in the first measurement. Of this initial sample, only 177 students were present in all four measurements (46.89% females, with ages ranging from 8 to 15 years). None of the students who agreed to participate dropped out of the study, but 290 of them did not attend school during the days of the second, third, or fourth measurement, so their information was not considered in the longitudinal analysis.

Participation was voluntary, and every lead teacher gave written informed consent after being previously informed of the main aspects of the study. Afterwards, on the first day of application, students were informed about the type of study that would be conducted and those who agreed to participate gave written informed consent. This study was approved by the Ethics Committee of the University of Concepción, Chile (ref. 04112016).

### 2.2. Instruments

The Hospital Anxiety and Depression Scale (HADS) [30]. In this study, we used a Chilean adaptation for children and adolescents [31]. This scale was administered to obtain depression and anxiety scores, taking both as a continuous variable without reference to a diagnostic category. This scale is composed of 14 items rated on a 4-point scale that provide anxiety (seven items) and depression (seven items) scores. According to data obtained by confirmatory factor analysis (CFA), a two-factor structure of anxiety and depression showed suitable indexes, with better fit indexes when two items (7 and 8) were removed (Chi-squared (X^2^) = 96.86 (*p* < 0.05), X^2^/df = 1.86 (df: degrees of freedom), Root mean square error of approximation (RMSEA) = 0.043, Goodness of fit index (GFI) = 0.993, Adjusted goodness of fit index (AGFI) = 0.987, Normed fit index (NFI) = 0.899, Non-normed fit index (NNFI) = 0.937, Comparative fit index (CFI) = 0.950), as observed in previous studies [32,33]. The internal consistencies reached in this study were α (alpha) = 0.75 for the anxiety subscale, and α = 0.65 for the depression subscale.

The Positive and Negative Affect Schedule (PANAS) [34]. This is a 20-item scale with a 4-point Likert format in which 10 items measure positive affect (PA) and 10 measure negative affect (NA). A CFA with the initial sample of this study obtained adequate coefficients for a two-factor structure (X^2^ = 328.72 (*p* < 0.05), X^2^/df = 2.02, RMSEA = 0.048, GFI = 0.990, AGFI = 0.989, NFI = 0.902, NNFI = 0.937, CFI = 0.947). Internal consistencies were α = 0.84 for PA, and α = 0.85 for NA. These coefficients were similar to those found in Chilean adolescent samples [35].

Rosenberg’s Self-esteem Scale (RSES) [36]. This scale provides a general self-esteem score through a 10-item scale with a 4-point Likert format. Given that there was no adaptation to the Chilean adolescent population, we conducted a new confirmatory factor analysis (CFA) of the initial sample of this study. Data supported the single factor structure of the global self-esteem model (X^2^ = 30.55 (*p* < 0.05), X^2^/df = 2.04, RMSEA = 0.047, GFI = 0.998, AGFI = 0.994, NFI = 0.980, NNFI = 0.981, CFI = 0.990), with better fit indexes when two items (1 and 8) were removed, as they showed a very low relationship with the rest of the items on the scale. In addition, internal consistency was appropriate (α = 0.84) and similar to the coefficient found by a previous study in a young Chilean sample [37].

The Multidimensional Scale of Perceived Social Support (MSPSS) [38]. This scale is composed of 12 items that measure three perceived social support scores: support from family, friends (i.e., peers), and significant others. A CFA obtained adequate fit coefficients for these three factors (X^2^ = 70.88 (*p* < 0.05), X^2^/df = 1.51, RMSEA = 0.046, GFI = 0.991, AGFI = 0.983, NFI = 0.949, NNFI = 0.974, CFI = 0.982). For the purpose of this research, only scores on support from family and friends were used. Internal consistencies were identical for both subscales (α = 0.86), in the range of the coefficients found with Chilean adolescent samples [39].

### 2.3. Procedure

The fieldwork took place once the Ethics Committee of the University of Concepción had authorized the study. Eight schools (three municipals, three private subsidized, and two private non-subsidized) among the total number of schools in Concepción were randomly selected and contacted to explain the study and invite them to participate in it. In exchange, each school would receive general feedback on mental health indicators. Substitute schools were also randomly selected in case there were any difficulties, but it was not necessary to resort to them. Once the schools had authorized the participation, an informed consent form was sent to all students’ parents from fourth to eighth grade. After collecting the informed consent forms, each school assigned one or more classrooms where the instruments would be applied. On the first day of application, students whose parents had authorized their participation in the study were informed about its aim and characteristics (i.e., voluntary and confidential) and asked to sign an informed consent form if they agreed to participate.

All instruments where administered by graduate students from the University of Concepción and monitored by the second author of this article. It took around 20 to 40 min to respond to all instruments.

### 2.4. Data Analysis

As regards descriptive statistics, means and standard deviations of the psychological variables measured were calculated for the initial early and middle adolescent sample. Data were also calculated for males and females. Relationship patterns between depression and anxiety scores and the remaining variables were assessed with the Spearman method. To analyze the stability of measurements, repeated-measures ANOVAs were applied, comparing the following interval periods: time 1 (initial), time 2 (one-month), time 3 (two-months), and time 4 (four-months). A step-by-step multiple regression analysis was used to explore the potential predictor variables of depression and anxiety scores. Scores on positive affect, negative affect, self-esteem, and support from family and friends were used as predictor variables for each time-period. The procedure was performed considering intervals of 1, 2, and 4 months.

## 3. Results

Descriptive statistics for psychological variables collected from the initial measurement are summarized in Table 1. As can be observed, we found a slightly higher score in depression compared to anxiety, higher levels of positive affect compared to negative affect, and higher perceived social support from family, compared to support from friends. No significant differences between males and females were found in any of those variables.

Regarding the relationship patterns between depression and anxiety levels and affectivity, self-esteem, and social support, Table 2 shows the Spearman correlation coefficients obtained. Interestingly, no relationship was found between depression and anxiety scores. The depression score showed significant and negative correlations with PA, self-esteem, and family/friend support, and a positive and significant correlation with NA. In this case, the relationship pattern was less robust than with PA. This pattern was observed in females and males. Yet, considering social support, females showed a higher negative correlation with family support, and males did with friend support. The anxiety score showed some differences with depression score correlation patterns—in this case, we found higher correlations with PA. Social support reached lower coefficients compared to depression level, both in males and females, and the correlation between anxiety level and support from friends did not reach statistical significance in males.

Regarding the stability analysis of depression and anxiety scores (and the rest of psychological variables measured) during the 1-, 2-, and 4-month intervals, we conducted a repeated-measures analysis of variance. Table 3 summarizes the coefficients found. As shown in the table, this analysis showed differences only in PA that were marginally significant, with a tendency to decrease at one month, and stabilized between the third and fourth measurements.

Taking into account the stability of the measures, a final group of analyses was carried out to determine the prediction of depression and anxiety scores. Three multiple regression analyses (step-by-step method) were conducted in order to predict the levels of both depression and anxiety scores at the 1-, 2-, and 4-month intervals. Because correlational analyses (Table 2) showed significant coefficients, PA, NA, self-esteem, family support, and friend support were taken as predictive variables, as they were assessed at the initial time-period. Given that males and females showed a similar pattern of relationships, we performed the analyses for the whole sample. The summary of these analyses is shown on Table 4.

The regression analysis for depression levels generated significant models in the 1-, 2-, and 4-month periods. The 1-month interval model included three steps (F = 23.56, *p* < 0.05) and explained approximately 30% of the variance (R^2^ = 0.28), revealing a function made up of PA in the first place, followed by family support and NA. The 2-month interval model also included three steps (F = 19.99, *p* < 0.05) and explained approximately 25% of the variance (adjusted R^2^ = 0.25), showing a function in which Initial PA was the best predictor of depression level, followed by self-esteem and family support. Finally, the 4-month interval model only included one step (F = 37.09, *p* < 0.05) and explained approximately 15% of the variance (adjusted R^2^ = 0.17), exhibiting a function in which Initial PA was a predictor of depression score. In all models, the predictor variable that remained constant was PA, which was negatively associated with depression levels.

As regards anxiety level, the three regression models were also significant. The 1-month interval model had four steps (F = 46.55, with *p* < 0.05) and explained approximately 50% of the variance (R^2^ = 0.51), indicating a function made up of NA in the first place and followed by PA. The 2-month interval model had three steps (F = 30.45, *p* < 0.05) and explained approximately 35% of the variance (R^2^ = 0.34), showing a function in which Initial NA was the best predictor of anxiety score, followed by friend support and self-esteem. Finally, the 4-month interval model had two steps (F = 29,311, with *p* < 0.05) and explained approximately 25% of the variance (adjusted R^2^ = 0.24), revealing a function in which Initial NA was the best predictor of anxiety level, followed by PA. In all models, the predictor variable that remained constant was NA, which was positively associated with anxiety levels.

## 4. Discussion

The purpose of this study was to identify the predictive value of a group of psychological variables that are frequently associated with emotional disorders (i.e., depression and anxiety) in adolescence, namely affectivity, self-esteem, and social support. To do so, an adolescent sample was followed for four months and assessed four times. These variables showed stability across those four assessment intervals (except for an interval in which positive affect tended to decrease). Moreover, correlation coefficients showed a close relationship between (1) affectivity, self-esteem, and social support, and (2) depression and anxiety scores. Higher coefficients were found between depression level and positive affect, and also between anxiety level and negative affect. These data were confirmed with regression analyses, since both affects were the best—and occasionally the only—predictors of depression and anxiety scores. Although the correlation coefficients between depression and anxiety scores were positive, they were low. Figure 1 shows the participation of affectivity (positive and negative), self-esteem, and social support (i.e., from family and friends), counting the times these variables participated in each of the three regression analyses performed.

The analysis of the predictability of depression and anxiety scores revealed that self-esteem and social support played an unstable role. Self-esteem played a marginal role, as its predictive ability was observed in two of the six regression analyses performed (and in the 2-month interval). As regards the theoretical explanation of the relationship between self-esteem and emotional disorders, self-esteem seems to have a reciprocal relationship with depression and anxiety levels [20] rather than show a vulnerability state: there were high and significant correlation coefficients between self-esteem, anxiety, and depression, but self-esteem had scarce participation in predicting these emotional problems.

Social support (i.e., from family and friends) participated in predicting emotional disorders, but only in the shorter intervals (1 and 2 months) and with a differential pattern—while family support predicted depression scores, friend support was more present in anxiety scores. This pattern is difficult to explain because existing data back the negative association between both family and peer support and emotional disorders, particularly depression level [24]. Interestingly, this differential role may be justified by a developmental perspective—that is, family support and friend support may have a different participation in depression or anxiety scores depending on their relevance at each developmental stage [26]. However, targeted studies are needed to explain this differential participation of social support. In addition, perceptual processes can be affected by emotional disorders (especially depression), so perceived social support may be affected as anxiety and depression scores increase.

Affectivity was the only variable present in all regression analyses. PA was found to predict depression scores at one month, two months, and four months, and was the first variable in the regression equation. NA played a similar role in predicting anxiety scores. These results are in line with the tripartite model [12]. This well-known model postulates that anxiety and depression share a positive relationship with NA and a negative relationship between depression and PA (identified as anhedonia). This model has received a broad empirical support, with different types of samples and at various age stages, including adolescence [40]. The weak relationship between depression and anxiety scores may challenge the appropriateness of the tripartite model, but these data have also been obtained by other studies, suggesting that depression and anxiety have different onsets [41]. Nevertheless, two statistical aspects may have altered the real relationship pattern between anxiety and depression scores: first, the internal consistency of the depression subscale was only moderate; second, two items of the HADS were removed to improve model fit. Both items were loaded on both subscales, and this decision may have affected the relationship between subscales.

Specifically in adolescents, low PA, which is mainly associated with depression, has also been found to be associated with anxiety, and particularly with social anxiety [39]. This is consistent with our data, in which anxiety level prediction was sustained for both NA and PA. This pattern of affectivity during adolescence has been supported in longitudinal studies, showing a stable presence of anhedonia and NA levels associated with emotional disorders. Yet, in the transition to adulthood, negative affect reflects a declining trajectory and a closer relationship between anhedonia and depression [42]. Moreover, impairments in reward-processing behaviors have been associated with depression and recovery from it [43,44]. These impairments can be appreciated in an altered brain reward-system circuitry and may reflect low PA, and ultimately, high NA [45,46].

Overall, our data supported the relevant role of affectivity as the best predictor of depression (PA) and anxiety scores (NA), according to the tripartite model. Self-esteem partly participated in predicting emotional problems, which calls into question its role as a predictor in favor of a reciprocal relationship with anxiety and depression levels. Social support had a more consistent relationship with emotional disorders, but its presence or absence, depending on the measurement interval, may suggest a contextual-situational role. Moreover, family support had a stronger relationship with depression scores, and friend support was more associated with anxiety scores. Taking these data as a whole, a hypothetical model can be developed, considering that there are “core” variables (such as affectivity) and “enhancer” variables (such as self-esteem and social support) in predicting emotional disorders. Yet, a specific design needs to be implemented to test this.

Regarding possible practical or clinical implications, our results suggest the need to consider the assessment of affectivity development from childhood to early adolescence as a secondary prevention strategy. If there is inadequate development, for example, with higher levels of negative affect and/or lower positive affect, psycho-educational programs can be implemented to reach a balanced affectivity process. In addition, social support (basically from family and friends) and the level of self-esteem can be explored, because inadequate—or lack of—social support and low self-esteem can increase the risk of appearance of both depression and anxiety processes when they are present along with unbalanced affectivity processes. Psycho-educational programs are likely to be particularly useful with the Chilean child-adolescent population, because of its tendency to have high rates of emotional disorders [4].

This study has several limitations, such as its sample size, which was representative in the initial measurement but was small to generalize the results obtained. The time intervals were also short for a longitudinal study. Likewise, the sample was recruited from a non-clinical population, so its clinical implications must be taken with caution. Given that different developmental stages can lead to different results, the middle childhood and adolescence age ranges used in this study are only a part of the adolescence spectrum.

Future research should consider larger samples—including a clinical sample—and time intervals that are long enough to make more robust predictions about depression and anxiety in adolescence. Furthermore, more complex models could be tested, introducing temperamental and environmental variables and especially variables related to the reward-processing system, because of their relationships with affectivity.

## 5. Conclusions

Depression and anxiety levels in an adolescent sample were best predicted by affectivity (i.e., positive affect in depression and negative affect in anxiety). Social support (i.e., from family and friends) seemed to play a contextual-situational role in predicting those emotional problems. Self-esteem exhibited a reciprocal relationship with depression and anxiety.

## Figures and Tables

**Figure 1 ijerph-17-06984-f001:**
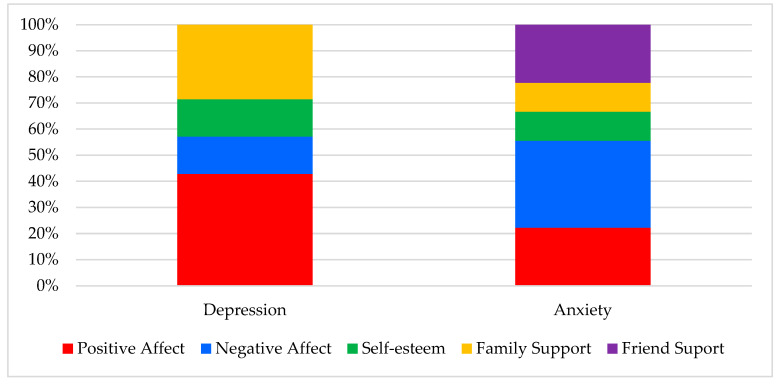
Differential participation of positive affect, negative affect, self-esteem, family support, and friend support in predicting depression and anxiety scores, taking into account the three multiple regression analyses.

**Table 1 ijerph-17-06984-t001:** Descriptive statistics for the total sample and male and female groups collected from the initial measurement.

	Mean	SD	Mean	SD	Mean	SD
	Total sample (*N* = 467)		Males (*N* = 230)		Females (*N* = 237)	
Depression	11.25	3.11	11.30	3.05	11.20	3.17
Anxiety	10.51	3.41	10.37	3.40	10.65	3.41
Positive affect	32.27	5.27	32.43	5.21	32.11	5.33
Negative affect	18.38	5.76	17.90	5.72	18.85	5.76
Self-esteem	24.37	5.13	24.66	4.99	24.08	5.26
Family support	12.80	3.32	13.08	3.01	12.53	3.58
Friend support	11.70	3.46	11.37	3.49	12.02	3.40

Note: SD = Standard deviation.

**Table 2 ijerph-17-06984-t002:** Spearman correlation coefficients between (1) depression and anxiety and (2) affectivity, self-esteem, and social support (*n* = 467).

	Anxiety	Positive Affect	Negative Affect	Self-Esteem	Family Support	Friend Support
Total sample						
Depression	0.07	−0.47 **	0.19 **	−0.37 **	−0.26 **	−0.30 **
Anxiety	---	−0.26 **	0.58 **	−0.35 **	−0.19 **	−0.08
Females						
Depression	0.09	−0.50 **	0.22 **	−0.37 **	−0.30 **	−0.26 **
Anxiety	---	−0.28 **	0.60 **	−0.31 **	−0.18 **	−0.14 *
Males						
Depression	0.05	−0.44 **	0.16 *	−0.36 **	−0.21 **	−0.34 **
Anxiety	---	−0.23 **	0.55 *	−0.38 **	−0.20 **	−0.03

Note: ** *p* < 0.01; * *p* < 0.05.

**Table 3 ijerph-17-06984-t003:** Multivariate analysis of variance for depression, anxiety, affectivity, self-esteem, and social support at the four time-periods (*N* = 177).

	T1		T2 (One-Month)		T3 (Two-Months)		T4 (Four-Months)		F	η2
Variables	Mean	SD	Mean	SD	Mean	SD	Mean	SD		
Anxiety	10.30	3.31	10.21	3.51	10.60	3.84	10.38	3.84	1.48	0.008
Depressive mood	11.72	2.94	11.62	3.22	11.57	3.36	11.42	3.19	0.63	0.004
Positive affect	32.36	4.97	31.89	5.56	31.63	5.85	31.68	5.65	2.76 *	0.015
Negative affect	17.37	5.15	17.71	5.72	17.54	5.87	17.37	6.01	1.16	0.007
Self-esteem	24.94	4.55	24.62	5.05	24.91	5.44	24.92	5.24	1.44	0.008
Family support	13.11	2.99	12.84	3.05	12.61	3.43	12.70	3.43	0.38	0.006
Friend support	11.65	3.45	11.49	3.60	11.53	3.65	11.46	3.43	2.15	0.012

F: Fisher’s F-test; η2: squared Eta; * *p* < 0.05.

**Table 4 ijerph-17-06984-t004:** Step-by-step multiple regression analyses of depression and anxiety scores in the four time intervals (*n* = 177).

	B	t	*p*	95% C.I. Lower/Upper
Depression: Predictor variables in the 1-month interval				
Constant	18.27	9.11	0.000	14.32/22.23
Positive affect	−0.17	−3.17	0.002	−0.268/−0.06
Family support	−0.26	−3.25	0.001	−0.42/−0.10
Negative affect	0.12	2.70	0.008	0.03/0.21
Depression: Predictor variables in the 2-month interval				
Constant	22.96	15.25	0.000	19.99/25.93
Positive affect	−0.17	−2.77	0.006	−0.29/−0.05
Self-esteem	−0.14	−2.20	0.029	−0.27/−0.02
Family support	−0.17	−1.99	0.049	−0.34/0.00
Depression: Predictor variables in the 4-month interval				
Constant	20.10	13.95	0.000	17.26/22.95
Positive affect	−0.267	−6.09	0.000	−0.36/−0.18
Anxiety: Predictor variables in the 1-month interval				
Constant	7.61	4.23	0.000	4.06/11.17
Negative affect	0.40	9.91	0.000	0.32/0.48
Positive affect	−0.17	−3.45	0.001	−0.26/−0.07
Friend support	−0.26	4.20	0.000	0.14/0.39
Family support	−0.15	−2.00	0.047	−0.30/0.00
Anxiety: Predictor variables in the 2-month interval				
Constant	7.98	3.43	0.001	3.38/12.57
Negative affect	0.31	5.47	0.000	0.20/.42
Friend support	−0.28	3.74	0.000	0.13/0.42
Self-esteem	−0.24	−3.56	0.000	−0.37/−0.11
Anxiety: Predictor variables in the 4-month interval				
Constant	11.44	4.73	0.000	6.67/16.21
Negative affect	0.27	4.90	0.000	0.16/0.38
Positive affect	−0.18	−3.11	0.002	−0.29/−0.06

Note: B = beta coefficient; t: Student’s t; p = probability; C.I. = confidence interval.

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
