# Peer review of "The Predictive Role of Affectivity, Self-Esteem and Social Support in Depression and Anxiety in Children and Adolescents"

_ijerph, 2020, doi:10.3390/ijerph17196984_

Round 1

Reviewer 1 Report

I commend the authors of this manuscript for having conducted the fascinating study described on the attached manuscript. With that being said, I here add comments on how to improve the quality of the manuscript.

Title:

First, the manuscript identifies the ages of the participants as ranging from 8 years to 16 years. Since age 8 and its proximal ages are considered childhood and not adolescence, I here suggest that the title of the article be changed to something along the lines of the following: “The Predictive Role of Affectivity, Self-esteem and Social Support on Depression and Anxiety in Children and Adolescents”. Likewise, I suggest the term “children” to also be incorporated throughout the manuscript anytime data from participants close to the age of 8 were to be alluded to on the manuscript.

Abstract:

On the part of the abstract addressing results, the use of the word “by” should be replaced by the word “between”. Later, on the first sentence of the abstract addressing conclusions, the word “presented” should be edited to reflect present tense, through the word “present” on both occasions that word appears on that part of the abstract. On the final occasion that this word appears, it should be followed by the word “in” ,to replace the word “for”. Also, on this same sentence the word “analyzes” should be changed to the word “analyses”, thereby allowing that sentence to now read as follows: “Taking into account 23 initial scores of affectivity, self-esteem and social support in predicting both depression and anxiety 24 scores at one-month, two-months and four-months intervals, positive affect was present in three 25 regression analyses, predicting depression scores; and negative affect was present in anxiety 26 scores.”

Introduction:

On the first sentence of the first paragraph of the introduction, the word “the” should appear before the words “life span”. On the introduction’s second sentence the words “point out rates” should be replaced by the word “range”. Similarly, on the introduction’s second sentence the word “the” should be appear before the term “most frequent disorders”.

On the initial sentence of the introduction’s third paragraph the word “important” should be replaced with the word “prevalent”.

On the initial sentence of the introduction’s fourth paragraph the word “relevant” should be replaced with the word “high”. Then, on the last sentence of the same paragraph the terms “self-concept-self-esteem” and “support/context” should be divided, in order to be properly expressed.

On the third sentence of the introduction’s fifth paragraph the term “steps of adolescence” should be replaced with the term “few years of the adolescence period”. Then, on the fourth sentence of the introduction’s fifth paragraph the word “being” should be replaced with the word “been”.

On the initial sentence of the introduction’s sixth paragraph the first use of the word “been” should be erased. The comma used on the subsequent sentence (second sentence of the sixth paragraph) should be erased. The third sentence of the sixth paragraph should have a colon or semi colon prior to the first paragraph (i).

The seventh paragraph’s second sentence of the introduction should add the word “if” inside its parenthesis, so it can read as follows: “(e.g., such as if they perceive their parents are using emotional manipulation strategies). On the same sentence the term “are found” should be replaced with the term “have been found to be”. Also, on the same sentence the term “the development of” should be added right before the first use of the term “emotional disorders” in this sentence. Finally, the use of the term “non-formal care” on this same sentence should be revised, in order to better explain what the authors what to express through that term. The seventh paragraph’s final sentence should have the word “the” erased, and the word “because” replaced by the word “by”.

On the initial sentence of the introduction’s final paragraph the term “is directed to the analysis of” should be replaced with the term “aims to address”. Also, on the final sentence of the introduction’s final paragraph, the word “moments” should be replaced by the term “time points”. Similarly, the final sentence of the introduction’s final paragraph should be changed, to now read in a manner similar to the following: “To properly study relationships between these constructs, a longitudinal study was conducted on an infant-youth sample for four months, with four measurement time-points”.

Materials and Methods

The initial sentence of the participant’s section initial paragraph should have the word “of” replaced by the term “around the city of Concepción in Chile”. The second sentence of the participant’s section initial paragraph should have the term “was composed by” replaced by the term “consisted of”. Similarly, the word “being” should be added right before the word “females” on the same sentence. Also, the period after the word “females” should be replaced with a comma, as a means to blend this sentence with the subsequent and final sentence. Blending these sentences should allow this final sentence to now read as follows: “This sample consisted of 467 children and adolescents, 50.75% of them being females, with ages that ranged from 8 to 16 years old.”

The initial sentence of the participant’s section third paragraph should be reworded to now read in a manner similar to the following: “Participation on this study was voluntary, as documented through an informed consent form, signed by every participant and their tutor/legal guardian”. Also, from here on forward, the term “informed consent” should be expressed in lower caps.

The final sentence of the paragraph describing the Hospital Anxiety and Depression Scale mentions “α = .65 for depression”. That α could be considered to be categorized as questionable. Thus, I would like the authors of this study address whether that may play a part on the absence of relationship between depression and anxiety reported on this study.

The first use of the word “with” on the final sentence of the paragraph describing the Rosenberg´s Self-esteem Scale should be replaced with the word “on”. Similarly, the second use of the word “with” on the same sentence should be replaced with the term “study conducted across a”.

The second-to-last sentence of the paragraph describing the Multidimensional Scale of Perceived Social Support should be changed to the following: “For the purpose of this research, only support from family and friends scores were used”.

On the first procedure paragraph, the word “offer” should be replaced with the term: “invite them”. The fifth sentence from this paragraph should be changed to now include a comma right after the word participation. Additionally, the subsequent sentence should be changed to now read as follows: “After informed consent forms were collected, each school assigned one or more classrooms 128 where the instruments would be applied”. Then, on the final procedure paragraph, the word “applied” should be replaced by the word: “administered”. Similarly, the word “to” should be added right after the word “respond” on the final sentence of the paragraph section.

On the data analysis section, the word “obtained” should be replaced by the word “calculated” on both occasions that word appears on this paragraph. Similarly, the word “these” should be erased from this paragraph. Also, the term “carried out with” on this paragraph should be replaced with the term “assessed through”. Finally, near the end of this paragraph there is a phrase that states the following: “all the variables of the present study, except for the same dependent variable (anxiety or depressive mood).” That statement needs to be changed, since the predictor and outcome variables should be identified here.

Results

The first sentence of the results section’s first paragraph should be reworded as follows: “Descriptive statistics for psychological variables collected during the initial sample appear summarized in table 1.” Also, a comma should be added right after the word “depression” on the following sentence.

The header of table 1 should be reworded, to now read as follows: “Descriptive statistics for total sample, and groups of males and females, collected during the initial sample.

The second sentence of the results section’s second paragraph should be reworded as follows: “It results relevant to notice an absence of relationship between depression and anxiety.” Also, on the subsequent sentence the word “social: should be replaced by the word “friends”. Finally, near the end of this paragraph one sentence reads as follows: “Anxiety presents a similar pattern.” That sentence needs to be revised, since the pattern for anxiety should be explicitly explained, in terms of whether it addresses the total sample, as well as with male and females.

The third paragraph of the results section uses the term “bordering probability”. With that term, do you mean to refer to “marginally significant”? If so, please use that term instead.

The first word of the results section’ fourth paragraph (“Taking”) should be changed to the term: “To take”.

Discussion:

Please explain in more detail the content of the discussion section’s second parenthesis “(but an interval of positive affect)”. Similarly, specify which specific variables are alluded to on the fourth sentence of the discussion section’s first paragraph, which states the following: “Also, correlation coefficients showed a close 210 relationship between those variables with depression and anxiety.” Finally, the final sentence of the discussion section’s first paragraph should be revised, so that its ideas can be expressed with more coherence.

Later in the discussion there is one paragraph composed of only 1 sentence, with that being the following: “Specifically for adolescence, low PA, being mainly associated with depression, has also been found to be associated with anxiety, especially with social anxiety [39], and this is coincident with our data, where anxiety prediction is sustained for both NA and PA.” That is incorrect, since a paragraph should at minimum be composed of at least two sentences. Also, from that sentence the word “coincident” should be replaced with the word “consistent”.

Finally, the first sentence of the discussion’s third-to-last paragraph should be reworded to now read as follows: “In general, our data supports the relevant role of affectivity as the best predictor of depression and anxiety scores, according to the tripartite model.”

Reviewer 2 Report

Summary

This paper investigates the predictive role of affectivity, self-esteem and social support on depression and anxiety in an adolescent sample. The authors found that positive affect was negatively associated with depression scores and negative affect was positively associated with anxiety scores.

The research question is relevant, however, there is a lack of argumentation for what the present study adds to the field. Moreover, as the authors recognize themselves, the time span is too short to really evaluate the predictive role of included variables on symptoms of anxiety and depression. Drop-out rates are not presented making it difficult to determine quality and generalizability of the results.

Major comments

  1. The research question is relevant, however, my concern is the study design: analysing the stability of depression and anxiety levels and the predicitve role of affectivity, self-esteem and social support would ideally require more than a 4 month follow-up. A justification of the choice of study outline and argumentation on the relevance of multiple data collection points during such a narrow time-span would be helpful.

  1. What does this study add to the field? Is it the combination of relevant risk factors that makes the study unique? As the authors mention, the included factors have all previously been shown to impact the risk of internalizing disorders.

  1. The participation rate needs to be better described. How were the 177 participants of the follow-up selected? Was there a difference between these 177 and the 290 who were not included in the follow-ups? Did all 177 take part in all the follow-ups? This is important for understanding representativeness and generalizability of the results.

  1. Authors identify relevant limitations, however, the possible consequences for the results should be discussed. The authors mention the small sample size, but representativeness in relation to the baseline population is not discussed.

Minor comments

Page 2 line 46. The authors claim that ”Despite relevant rates of emotional disorders, risk factors remain unclear.” Please specify what you mean. In contrast, the abstract states that ”these variables belong to those psychological variables that are frequently associated with the prediction of emotional disorders”. Risk factors for depression and anxiety are well documented in the literature, (whereas the mechanisms of action are less clear). It is also unclear how the authors categorize risk factors; ACEs, environmental, normal developmental, biological, temperamental, psychosocial, psychological, socio-familial? These are all relevant variables, however, restructuring of the paragraphs concerning risk factors would increase clarity of the introduction.

Page 2, line 79. ”Infant” is not the correct term for 8 year olds.

Page 2, line 83. How was representativity determined (which factors)?

Page 3, Instruments. The authors use confirmatory factor analysis, excluding items for better fit indexes. How does that impact comparability with previous studies using the same instruments?

Page 6, line 214 and page 7, line 238. The low correlation between anxiety and depression scores is an unexpected finding which the authors could elaborate on more. Are there any possible methodological explanations to this finding?

Page 7, lines 227-230. Please clarify this statement about the differential role of family support and peer support in relation to the findings of van Harmelen et al. 2016.

General:

The manuscript is in need of language editing throughout. (For example: Abstract, line 21: ”with not significant differences”, page 2, line 59: ”have being”, page 4, line 147: ”according initial sample”, page 3, line 156: ”A first data is the…” page 5, line 180: ”correlational analyzes shown…”, page 7, line 225-226: ”This pattern has a difficult explanation”, page 7, line 251: ”according tripartite model.”, but these are just a few examples).

This is an interesting paper discussing a relevant topic but in my opinion, it is not yet ready for publication in its present form.

Reviewer 3 Report

The paper aims to identify inter and intra-subject stability of depression and anxiety levels (measured by the HADS scale), considering affectivity, self-esteem and social support levels. Affectivity, self-esteem and social support were identified to predict both depression and anxiety scores at one-month, two-months and four-months intervals, positive affect was presented in three regression analyses, predicting depression scores; and negative affect was presented for anxiety scores.

Detailed comments, concerns and suggestions

  1. In the introduction, and since the study was conducted in Chile, it would be important to have some contextual background about why conducting this study would matter for Chile. Why is it important to do such study there?
  2. In the participants section, the authors should provide more information concerning the population under study. “An initial representative sample” –please specify what kind of representativeness this sample has and why, what larger group is this sample adequately replicating and how the authors calculated the sample size. Why 467 adolescents? How many you intended to reach, is there a concern for self-selection of those that answered? Was the sample stratified by age and gender? (and other criteria).

The authors also state that “From that initial sample, 177 were assessed” – retention rate should be stated and also what happened to the initial sample of participants, this is not clear.

  1. Instrument section

 In this study a Chilean adaptation of the Hospital Anxiety and Depression Scale (HADS) for children and adolescents was used. This adaptation study hasn’t been published yet. The authors provided confirmatory factor analysis (CFA) and internal consistencies data from previous studies [31-32]. I believe the reader would like to be informed about the HADS scale behaviour in the Chilean youth population as well.

Also, regarding the HADS, this scale was created with the intent of being a tool for the detection of anxiety and depression levels in people with physical health problems. Why did the authors choose this tool for a school environment?

The HADS scores were considered the outcome measures in the study. The authors use different words along the text to define the outcome they are studying: Depression/Anxiety disorders? Mood disorders? emotional disorders? Or is it Depression / Anxiety levels? This needs to be further specified taking into account that the HADS scale only determines the levels of anxiety and depression and is not a diagnostic instrument.

It would also be beneficial if the authors could specify in the Instruments section exactly what is the measure each instrument is providing.

There is quite a lot evidence that gender and age are important variables when studying mood disorders. Did the authors consider these variables altogether?

  1. Discussion and conclusion

I am not sure about the originality of this study beyond the already existing consensus in this research area about the predictive effects of affectivity, self-esteem, and social support on the onset of mood disorders – which is also stated in the background of the manuscript.

I would recommend to revise the paper to bring out your main findings more strongly particularly on how this research adds new knowledge to this research area.

Reviewer 4 Report

In this study, the authors have examined the relationship between perceived affect (self-esteem, social support, and overall positive/negative affect) and depression/anxiety. The premise of the study is interesting, and the study design is sound. The authors found correlation between anxiety/depression and self-report of esteem and perceived social support. The data are interesting and add to the understanding of mental illnesses. Following are my comments:

  • However, the interpretation of the data is a bit to strong. The design of the study does not test predictability. At most, the study is examining correlation. It is possible that individuals with depression perceive low social support and self-esteem, instead of the opposite.
  • Data presentation with figures would be helpful instead of tables.

Round 2

Reviewer 3 Report

The author’s improved the manuscript substantially, yet there are still some minor concerns to be considered:

The authors responded that “Now, our data are referred as depression scores or anxiety scores, and we referred to depression disorder or anxiety disorder referred to publications with participants with those diagnostic label. Those terms have been rewording in text.” However, inconsistencies still persist. Here are some examples of places where the inconsistencies are present:

Pag 2. Line 86 “prediction model for depression and anxiety in child-youth population” – Again the study cannot provide evidence for a prediction model for a diagnostic label. The depression and anxiety scores or levels concept, needs to be constant all over the text.

Pag 4. 170 All paragraph “ It results relevant to  notice an absence of relationship between depression and anxiety. Depression obtained significant and negative correlations with PA, self-esteem and family/friend support, and a positive and significant correlation with NA. In this case, relationship pattern is less robust than with PA. This pattern can also be observed both for females and males, but, taking into account social support,  females show a higher negative correlation with family support, and males with friends’ support. Anxiety presents some differences with depression correlation patterns: In this case, higher correlation is found with PA. Social support attained lower coefficients compared with depression both for males and females, and the correlation between anxiety and friend support does not reach statistical significance, in males”

The depression and anxiety scores or levels concept, needs to be constant all over the text.

In the discussion

It would be important to state a summarized version of  the strengths that the authors replied “ We think our results can be understanding as a modest contribution to knowledge in this field. We agree there are many studies, with different designs, dealing with the prediction of emotional disorders in adolescent samples. Then, what we are adding to knowledge in this field? We think we are adding a modest contribution, in three aspects: (i) we take into account three groups of variables, more frequently associated with emotional disorders, but taking simultaneously them. (ii) We study these variables in a longitudinal design. We can understand these designs require more than four months, but our contribution is related with the four measurements in four time periods, to add support to a consolidated prediction model. And (iii) we think our data support a modest contribution to knowledge in the sense that there can be “nuclear” variables (as affectivity) and “enhancer” variables (as self-esteem and social support), in predicting emotional disorders.”

Also, it would be beneficial to reinforced in the discussion that this paper is especially relevant for Chile since “In Chile, the child-adolescent population presents high percentages in the prevalence of mental disorders” and also to help inform possible public health measures? Or other recommendations? Maybe add something related to this in the end?
